# Sea Port SO₂ Atmospheric Emissions Influence on Air Quality and Exposure at Veracruz, Mexico

Gilberto Fuentes García [1],*, Rodolfo Sosa Echeverría [1], Agustín García Reynoso [1], José María Baldasano Recio [2,3], Víctor Magaña Rueda [4], Armando Retama Hernández [5] and Jonathan D. W. Kahl [6]

1   Sección de Contaminación Ambiental, Ciudad Universitaria, Instituto de Ciencias de la Atmósfera y Cambio Climático, Universidad Nacional Autónoma de México, Ciudad de México 04510, Mexico
2   Barcelona Supercomputing Center-Centro Nacional de Supercomputación (BSC-CNS), Department of Earth Sciences, Nexus II Building, Jordi Girona 29, 08034 Barcelona, Spain
3   Department of Engineering Design, Universitat Politècnica de Catalunya (UPC), Av. Diagonal 647, Planta 10, 08028 Barcelona, Spain
4   Clima y Sociedad, Ciudad Universitaria, Instituto de Geografía, Universidad Nacional Autónoma de México, Ciudad de México 04510, Mexico
5   Independent Researcher, Ciudad de México 11800, Mexico
6   School Freshwater Science, University of Wisconsin-Milwaukee, Milwaukee, WI 53201, USA
*   Correspondence: fuenbeto@me.com

**Abstract:** In this work, we identify the current atmospheric sulfur dioxide emissions of the Veracruz port, an important Mexican seaport experiencing rapid growth, and its influence on the surrounding areas. Sulfur dioxide emissions based on port activity, as well as meteorology and air quality simulations, are used to assess the impact. It was found that using marine fuel with low sulfur content reduces emissions by 88%. Atmospheric emission estimates based on the bottom-up methodology range from 3 to 7 Mg/year and can negatively impact air quality up to 3 km downwind. After evaluating different characteristics of vessels in CALPUFF, it was found that maximum sulfur dioxide concentrations ranging between 50 and 88 µg/m³ for a 24-h average occurred 500 m from the port. During 2019, five days had unsatisfactory air quality. The combination of a shallow planetary boundary layer, low wind speed, and large atmospheric emissions significantly degraded local air quality.

**Keywords:** air quality modeling; coastal area; impact on air quality; atmospheric emissions; air pollution; fuel consumption; sulfur content in marine fuel





## 1. Introduction

Emissions from maritime transport have a negative influence on the air quality affecting the population living near coastal areas [1]. In addition to local environmental impacts, ship emissions degrade air quality regionally [2–11] and globally [12–16] and adversely affect public health [17,18].

Exposure to atmospheric emissions from port activity is mainly due to the combustion process that takes place in the main engines (ME) and auxiliary engines (AE) from ships during maneuvering and docking activities [19–22].

It has been estimated that ship traffic causes approximately 60,000 premature deaths each year globally [14]. Long-term exposure is associated with mortality from cardiopulmonary disease and reduced life expectancy [23–25], and short-term exposure is also associated with mortality as well as hospitalization for cardiovascular and respiratory disease [26,27].

Given these considerations, the study of port emissions impact is increasingly relevant, given the global increase in fossil fuel consumption [28]. Thus, the International Maritime Organization (IMO) already has a plan for the long-term reduction of the sulfur content of marine fuel [28–31]. However, these reductions will not significantly reduce particulate emissions [32–38] even if the global fleet switches to low-sulfur fuel.

A global impact assessment estimated that widespread use of 0.5% mass by mass (m/m) low sulfur fuel by ships in coastal regions would prevent approximately 33,500 premature deaths annually [39]. A health benefits assessment of introducing an Emissions Control Area (ECA) in the Marmara Sea and Turkish Straits suggested that such intervention would avoid approximately 15 premature deaths annually in that region, which has a population of approximately 20 million [40]. The delay in implementation from 2020 to 2025 would contribute to more than 570,000 additional premature deaths compared to on-time MARPOL Annex VI implementation [41,42].

According to regional ship emission inventories [43,44], previously significant uncertainties in the estimated emissions of global ship traffic have decreased during the last half decade [45,46]. To reduce the uncertainty of atmospheric emissions, one method for estimating regional atmospheric emissions was investigated by [47]. The method consists of taking sample ships considering the ship's density, typology, and engine power based on the Automatic Identification System (AIS) for exhaust emissions.

In Mexico, recent studies [48–50] represent a starting point for the elaboration and updating of Atmospheric Emissions Inventories for maritime activity. They considered two traditional methods of identifying the atmospheric emission levels of the main pollutants and greenhouse gases (GHG) produced during the combustion process of different ship types. According to [51], atmospheric emission levels could increase in the coming years because Mexico depends on fossil fuels for electricity generation. Also, it has been reported that by switching fuels in some port sectors in Mexico, there would be a reduction of 1 to 6% for nitrogen oxides ($NO_x$), 50 to 70% for particles (PM), and >99% for sulfur oxides ($SO_x$) [52].

To identify the ground-level concentration pollution to which the population is exposed, and to ensure good air quality in the future, it is necessary to apply meteorology and air quality models [38,44,53–65]. Application of such models includes consideration of emission sources of uncertainties and model limitations, including model resolution, limitations due to the complexity of air quality assessment, meteorological data quality, and emissions inventory consistency [66,67].

Global warming and air pollution from burning fossil fuels (coal, oil, and gas) are closely linked. Fossil fuel combustion has a direct effect on human and environmental health, and an indirect effect on generating GHG [68,69]. However, shipping activities play an important role in global exhaust emissions and in emission projections for $SO_2$ and $NO_x$. In 2030 these are projected to be approximately 4000 and 6000 Gg, respectively [70]. According to the third GHG report released by IMO in 2014, carbon dioxide ($CO_2$) emission from international shipping activities in 2012 was 796 Tg, accounting for 2.2% of global emissions. Ship GHG emissions are expected to increase by 50–250% between 2012 and 2050 [71]. The need to reduce the level of atmospheric emissions does not exclude the use of fossil fuels, but it does require considering low sulfur content and fossil fuel quality. The use of carbon sequestration technology is necessary and is expected to result in emissions reductions of 16% per year by 2050 [72].

Total shipping $CO_2$ emissions increased from 910 Tg to 932 Tg (+2.4%) from 2013 to 2015. International shipping emissions increased by 1.4%, domestic shipping emissions increased by 6.8%, and fishing emissions increased by 17%. In 2015, total shipping emissions accounted for 2.6% of global $CO_2$ emissions from fossil fuel use and industrial processes. International shipping contributed the most, representing about 87% of total $CO_2$ emissions from ships each year [31,73–77].

Currently, $SO_2$ emissions from the port sector are a global concern because emission levels have caused a deterioration in materials, monuments, and human health. Ships are the primary source of port emissions of $SO_x$ and $NO_x$, accounting for 71% and 58% of the totals, respectively, emphasizing the importance of reducing ship emissions [78–80]. It has been reported that two main fuel types are used in port sectors. These are Residual Oil (RO) and Marine Distillate (MD). RO has a sulfur content of 2.7% m/m. The sulfur content of RO used in ships globally is typically in the range of 2.0% to 3.5% m/m, with a global

average of around 2.6% m/m [31]. In some markets in the Middle East and Asia, RO of significantly higher sulfur content dominates [81].

$SO_2$ interacts with other species in the atmosphere, converting to sulfates. With humidity and oxidizing agents, it may be converted to sulfuric acid, a major component of the acid rain phenomenon [70,82–84]. Its atmospheric half-life is hours [85,86].

Atmospheric emissions estimation in port is carried out through the bottom-up or top-down method depending on the information available regarding the ship type and fuel consumption during the maneuvering and docking intervals daily. Both methods have advantages and disadvantages, but they represent significant advances for generating emissions inventories in ports [15,16,87–92]. This activity is important because it represents the input data for the use of air quality models, allowing the identification of the environmental concentration level to which the population is exposed and its comparison with official air quality standards.

To reduce the uncertainty associated with the emissions estimation from ships during the cruise, maneuvering, and hoteling positions, integration of AIS into the bottom-up system is required [93–96]. The Ship Traffic Exhaust Emission System (STEAM) is based on AIS. It represents a crucial advance in identifying the impact on air quality by $SO_x$, $NO_x$, carbon monoxide (CO), $CO_2$, and particles due to global ship movement [93,97,98]. The difference between STEAM and other methods in determining ship emission is approximately 20% [99].

The CALPUFF and AERMOD models have been previously utilized to determine air quality impacts from maritime activities [56,100–102]. A variety of information regarding the technical aspects of ship stacks is available: height, diameter, temperature, and gas exhaust velocity [103–108]. These parameters are crucial for determining the environmental concentration of atmospheric pollutants, integrating the level of atmospheric emission, and meteorology considering the location of the ships during the docking position.

According to the literature review, the objective of this study was to identify areas experiencing negative air quality impacts (high 24 h $SO_2$ concentrations) caused by emissions from combustion within the engines of ships arriving at the port. The CALPUFF dispersion model version 5.8 [109] was used to determine the ambient concentrations by using (1) emission data daily based on the bottom-up system [110], (2) technical data of ship stacks: physical height, internal diameter, velocity and temperature of exhaust gases [103–107], (3) in-situ and reanalysis weather information processed in the Weather Research and Forecasting model (WRF) [111], and (4) comparison of the environmental exposure level according to the results of the CALPUFF model and the standard reference concentration [112,113]. Twelve events were selected to identify the maximum concentration to which the population was exposed by comparing the level of environmental concentration recorded at the ambient air quality monitoring station with the concentration estimated according to the CALPUFF model during 2019. The use of the cluster system of the HYSPLIT model [114,115] was utilized to identify 120-h backward trajectories from the study area from 2010 to 2020. This allowed the selection of events for the air quality simulation, as well as the analysis of the $SO_2$ measurement database from 2018 to 2020.

The availability of official information on the movement of ships daily made it possible to perform this study for the port of Veracruz, one of the most important ports in Mexico located on the Gulf of Mexico.

## 2. Port of Veracruz

Ship typology, GT, spent time in a berthing position, and frequencies are shown in Table 1 from 2018 to 2020. There was a decrease in ship frequency in 2020 due to the COVID-19 pandemic. However, Container ships are widely represented in the port system, with 500 ships per year. The GT level is around 60,000 for Containers, followed by RoRo Cargo, indicating that they contribute higher atmospheric emissions than other vessels. Docking times for Container Specialized are less than a day. However, its GT level is high, meaning the atmospheric emission level will also be higher. Likewise, emissions from RoRo Cargo are influenced by high GT and docking times of approximately two days, indicating higher

emissions. For the rest of the vessel types, the same situation does not exist because the GT level is less than 30,000, and port frequency is less than 20%. The highest frequency occurs in RoRo Cargo with 22%, while Container (both Specialized and Non-Specialized) is 34%.

**Table 1.** Typology of ships in the Veracruz port system.

| Type of Ship | Type of Cargo | Ships Calling 2018 | 2019 | 2020 | GT Average 2018 | 2019 | 2020 | Hoteling, Time (h) 2018 | 2019 | 2020 | Frequency 2018 | 2019 | 2020 |
|---|---|---|---|---|---|---|---|---|---|---|---|---|---|
| General Cargo | General | 340 | 358 | 296 | 27,298 | 28,979 | 30,672 | 85 | 88 | 75 | 11% | 11% | 12% |
| RoRo Cargo | Vehicles | 267 | 230 | 178 | 56,373 | 57,407 | 57,329 | 42 | 45 | 42 | 22% | 22% | 22% |
| Dry Bulk | Mineral | 172 | 125 | 148 | 20,343 | 20,314 | 19,182 | 102 | 92 | 94 | 8% | 8% | 8% |
| | Agricultural | 228 | 236 | 223 | 24,999 | 24,589 | 25,284 | 170 | 155 | 158 | 10% | 9% | 10% |
| Container | Non-Specialized | 171 | 168 | 110 | 28,441 | 27,031 | 22,486 | 21 | 21 | 16 | 11% | 10% | 9% |
| | Specialized | 456 | 494 | 532 | 59,293 | 60,435 | 59,153 | 19 | 21 | 17 | 23% | 23% | 23% |
| Liquid Bulk | Fluids | 172 | 177 | 186 | 11,757 | 13,807 | 14,479 | 32 | 54 | 61 | 5% | 5% | 6% |
| | Hydrocarbons | 174 | 208 | 188 | 29,093 | 28,367 | 26,839 | 70 | 88 | 94 | 11% | 11% | 11% |
| | Total | 1980 | 1996 | 1861 | 257,597 | 260,929 | 255,424 | 541 | 564 | 557 | 100% | 100% | 100% |

The locations of air quality (AQMS) and meteorological monitoring stations and dock positions are shown in Figure 1. There are 21 berthing positions at "Bahía Sur" and 5 berthing positions in operation at "Bahía Norte". AQMS and meteorology were located at 19.22 north (N) and 96.16 west (W). Background concentrations prior to the port expansion were measured at this location. Air quality impact was determined through this AQMS for "Bahía Norte" and "Bahía Sur" port activity. The selection of the measurement site was based on a meteorological study that identified the prevailing winds in the area and applied quality assurance (QA) and quality control (QC) protocols [116,117].

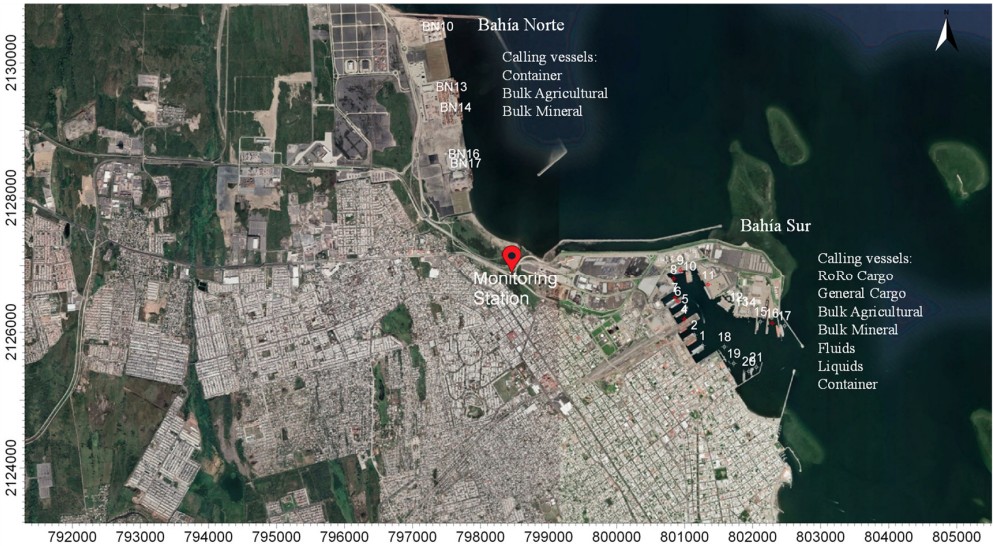

**Figure 1.** Port of Veracruz. Terrain design from WebGIS CALPUFF model.

The SO$_2$ concentration and meteorological parameters were obtained for the period from January 2018 to December 2020, with data every 10 min using an automatic SO$_2$ sampler, the equivalent method, and Vintage Pro equipment for recording surface meteorological data (wind speed and direction, humidity, temperature, solar radiation, pressure).

Wind direction and wind speed frequency are shown in Figure 2 from 2018 to 2020 at the AQMS. In summer, the prevailing winds are from the northeast (NE) and east (E), with a frequency of 10 to 20%. In autumn and winter, the dominant wind direction is N, with wind speeds from 6 to 22 m/s. Southeast (SE) winds were common in the spring season. Overall, The N and ENE winds were the most frequent from 2018 to 2020. These

characterize the sea breeze circulation. E and ESE winds were also common and represent airflow from the location of "Bahía Sur". NW, NNW, and N winds indicate airflow from the "Bahía Norte". The remaining wind directions indicate the contribution of $SO_2$ from other emission sources close to AQMS.

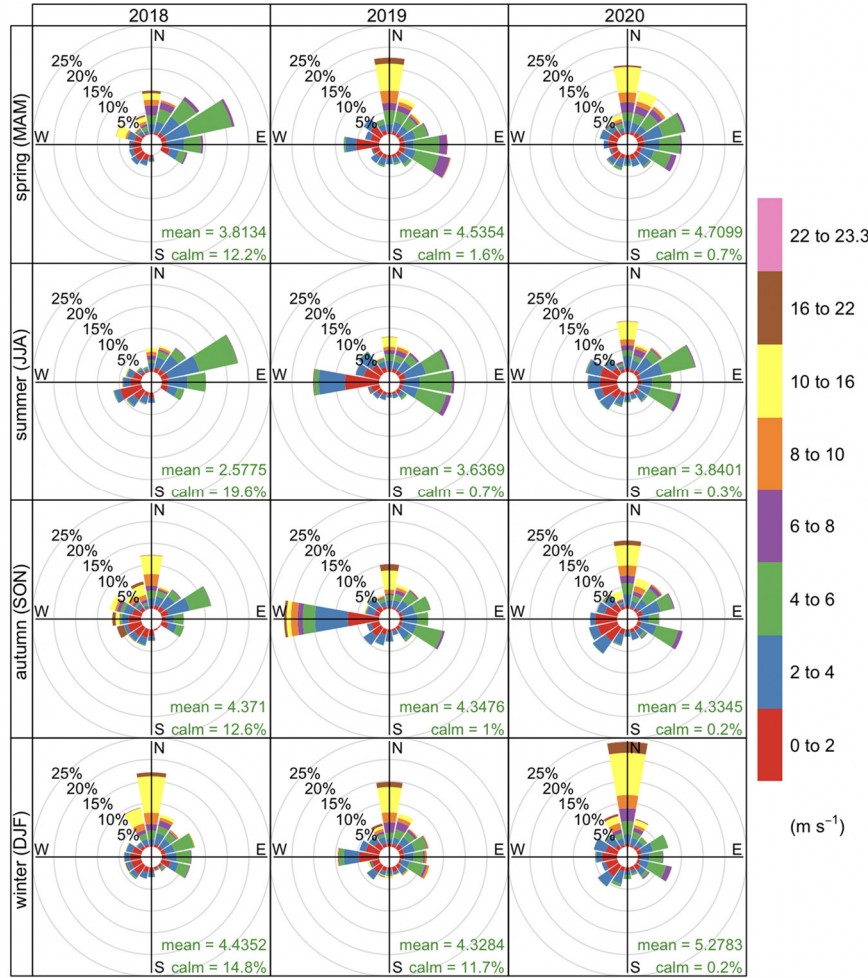

**Figure 2.** Seasonal wind rose at AQMS, 2018 to 2020.

Eventually, AQMS recorded measurements of $SO_2$ when the wind blew from N and E can provide information about possible impacts from port emissions.

## 3. Methodology

### 3.1. Atmospheric Emission

The Veracruz port system database from 2018 to 2020 [118] was downloaded to identify the annual activity of the port, taking 2019 as a base year for the development of this study. Information was available on the type of vessels, gross tonnage (GT), and the time of stay, meeting the requirements to estimate atmospheric emissions daily. These emissions were then used to identify ambient concentration levels using the CALPUFF model.

The daily emissions due to ship movement in port were determined using the bottom-up method [110] for the maneuvering and docking position. A sulfur content in marine fuel of 3.5% m/m is assumed because, in 2019, this was the value recommended by the IMO globally. The bottom-up method considers detailed ship information regarding ME and AE, power for both engines, GT, and the time spent in maneuvering and docking positions. The broadcast of the bottom-up system is shown in Equation (1), indicating that in the maneuvering position the ME and AE are in operation, while in the docking position only the AE is in operation. To determine the ME power, it was necessary to relate the GT by

ship type [119]. Likewise, the AE power was determined using the ratio of AE/ME and the ME power.

$$E = E_{maneuver.} + E_{berthing} \tag{1}$$

$$E = t_{maneuver.} * [(P_{ME} * LF_{ME} * EF_{ME}) + (P_{AE} * LF_{AE} * EF_{AE})] + t_{berthing} * (P_{AE} * LF_{AE} * EF_{AE})$$

where:

$E$: Total emission, g

$E_{maneuver.}$: Atmospheric emission in maneuvering position, g

$E_{berthing}$: Atmospheric emission in berthing position, g

$t_{maneuver.}$: Time spent in maneuvering position, h

$t_{berthing}$: Time spent in berthing position, h

$P_{ME}$: ME power, kW

$P_{AE}$: AE power, kW

$LF_{ME}$: Load factor of ME for each navigation phase

$LF_{AE}$: Load factor of AE for each navigation phase

$EF_{ME}$: Emission factor of ME for each navigation phase, $\frac{g}{kWh}$

$EF_{AE}$: Emission factor of AE for each navigation phase, $\frac{g}{kWh}$

A value of 20% for the load factor has been considered for all ships except for the Liquid Bulk, which utilizes a load factor of 40% in the hoteling phase for AE [6,110,120].

Parameters to determine $SO_2$ atmospheric emission are shown in Figure 3. According to the typology of ships in the port of Veracruz, we have considered the time spent in maneuvering and berthing positions, frequency, GT, ME, and AE power to identify their distribution on an annual average from 2018 to 2020. AE power is less than ME power because the ship propels through the ME, while the ship operation (loading and unloading of merchandise) corresponds to the AE. Atmospheric emissions are higher in the docking position because the frequency, stay times, and GT are important parameters for the atmospheric emission, contrary to the maneuvering position where two engines are in operation, but the time in this position is approximately 1 h. The combination of arrival frequency, GT, and docking time is the key to the atmospheric emission level. Considering the frequency of Specialized Container (23%) with GT above 60,000 and shorter stay time compared to other vessels, the highest atmospheric emission will be obtained, followed by RoRo Cargo. The lowest atmospheric emissions will come from Bulk Dry (Mineral) and Liquid Bulk (Fluids) because they have a GT between 13,000 and 20,000 with residence times greater than 24 h, but their frequency is 6 to 8%.

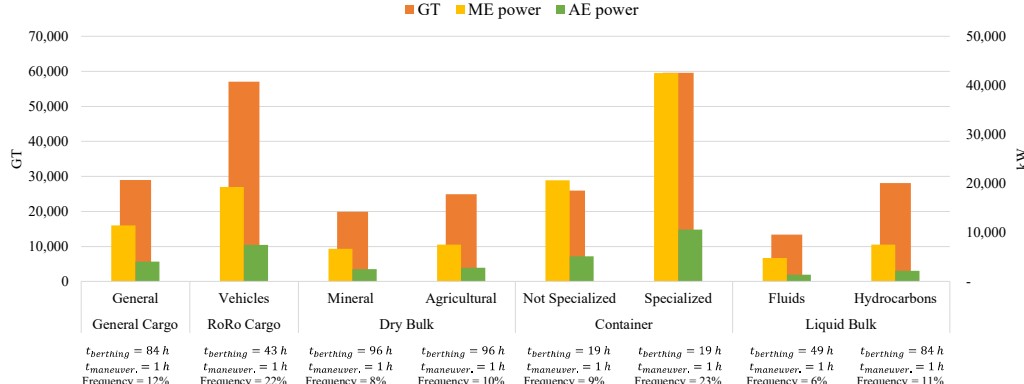

**Figure 3.** Parameters to determine atmospheric emissions by type of ship.

The emission factor $\left( \frac{kg_{SO_2}}{Mg_{fuel}} \right)$ to determine the level of atmospheric emissions by $SO_2$ in the 20s, where "s" represents the sulfur content percentage in marine fuel (%). During

the combustion process in the ME and AE, the chemical reaction (2) occurs, indicating that all S produces $SO_2$, assuming a conversion of 100%:

$$S + O_2 \rightarrow SO_2 \tag{2}$$

The $SO_2$ emission factor is obtained using Equation (3) considering the sulfur content, the molecular mass of S and $SO_2$ corresponding to (2), and 1 Mg of marine fuel as a calculation basis:

$$EF_{SO_2}\left(\frac{kg_{SO_2}}{Mg_{fuel}}\right) = \frac{1kgS}{100kg_{fuel}} * \frac{1000kg_{fuel}}{1\ Mg_{fuel}} * \frac{64kg_{SO_2}}{32kgS} = 20s \tag{3}$$

To relate the emission factor of Equation (3) to the operation of the ME and AE in maneuvering and docking position in accordance with Equation (1), the specific fuel consumption (SFC) is included according to engine type and vessel fuel usage. For example, if the sulfur content is 3.5% and the emission factor corresponds to 70 $\frac{kg_{SO_2}}{Mg_{fuel}}$, then it is necessary to consider the *SFC* for the type of vessel, marine diesel oil (MDS), the main engine $\left(\frac{223g_{fuel}}{kWh}\right)$ and AE $\left(\frac{217g_{fuel}}{kWh}\right)$. The emission factor is indicated in Equations (4) and (5)

ME:

$$EF_{SO_2} = \frac{70kg_{SO_2}}{Mg_{fuel}} * \frac{223g_{fuel}}{kWh} * \frac{1000g_{SO_2}}{kg_{SO_2}} * \frac{1kg_{fuel}}{1000g_{fuel}} * \frac{1Mg_{fuel}}{1000kg_{fuel}} = 15.61\frac{g_{SO_2}}{kWh} \tag{4}$$

AE:

$$EF_{SO_2} = \frac{70kg_{SO_2}}{Mg_{fuel}} * \frac{217g_{fuel}}{kWh} * \frac{1000g_{SO_2}}{kg_{SO_2}} * \frac{1kg_{fuel}}{1000g_{fuel}} * \frac{1Mg_{fuel}}{1000kg_{fuel}} = 15.19\frac{g_{SO_2}}{kWh} \tag{5}$$

*3.2. Air Quality Modeling and Meteorology*

A region of 15 km by 10 km was considered for modeling air quality. On it, the meteorological and atmospheric emissions information was processed in a mesh with 0.5 km grid size resolution. The terrain is shown in Figure 4. The terrain has elevations up to 50 m above sea level (masl) at 5 km from AQMS. The use of ring-type receptors was applied to identify the concentration level at 3 km from the source. Each ring has an annular distance of 0.5 km from the other. The land use and topography were downloaded from the WebGIS-CALPUFF model.

The input data used for modeling are (1) the ship's location in their docks, considered as emission point sources, (2) physical characteristics of the stack (diameter and height), (3) temperature and gas exhaust speed, (4) daily emissions, and (5) in-situ and reanalysis meteorology.

Because there is varying information about ship stack dimensions, gas exhaust temperatures, and velocity, we decided to use the information published by different authors (Table 2). Depending on the ship type or category, there may be more than one stack for exhaust gases. However, acquiring precise information on the number of stacks by ship type is complicated; therefore, an equivalent single stack is assumed for the exhaust gases [70–74].

**Table 2.** Characteristics of the point emission sources integrated at CALPUFF model.

| Reference | Height, m | Diameter, m | Exhaust Gas Velocity, m/s | Exhaust Gas Temperature, K |
|---|---|---|---|---|
| [103] | 36.5 | 1.5 | 5.0 | 373.0 |
| [104] | 20.0 | 0.8 | 25.0 | 540.0 |
| [105] | 40.0 | 1.0 | 10.0 | 573.0 |
| [106] | 44.0 | 0.5 | 7.5 | 583.0 |
| [107] | 30.0 | 0.5 | 20.0 | 573.0 |

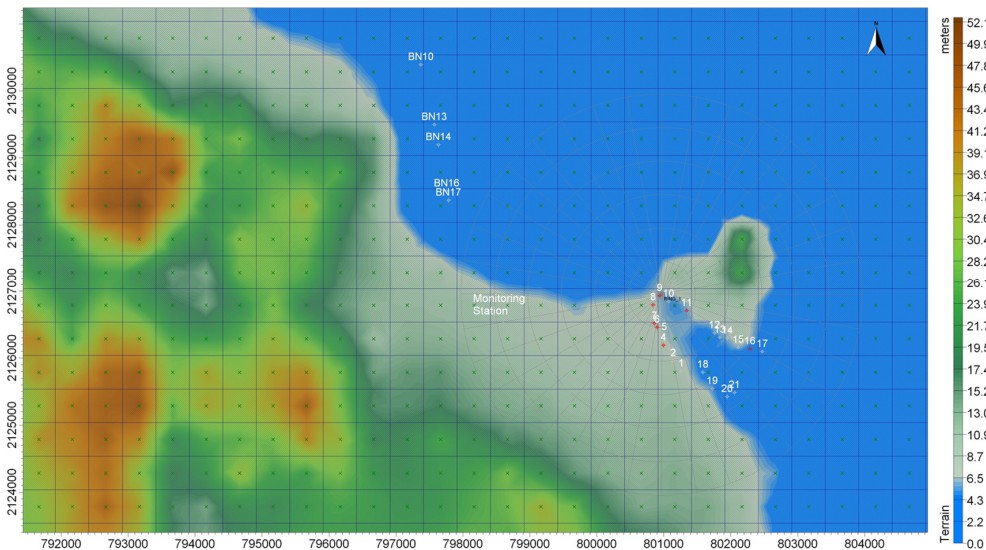

**Figure 4.** Air quality modeling area from the WebGIS CALPUFF model.

The WRFv3.9 model was used to generate hourly meteorological information. The initial and boundary conditions came from the Global Data Assimilation System (GDAS-0.25 degree) every six hours, downloaded directly from the National Centers for Environmental Prediction [121]. The Global Tropospheric Analysis and Forecast Grid were used. Parameters include surface pressure, sea level pressure, geopotential height, temperature, sea surface temperature, soil values, ice cover, relative humidity, u- and v- winds, vertical motion, vorticity, and ozone.

Three nesting domains were considered: the first domain had a dimension of $100 \times 100$ cells (27 km grid spacing) with a 10-min resolution for land use and topography. The second was $232 \times 190$ (9 km grid spacing) with a 5-min resolution. The third was $106 \times 106$ with 3 km spacing and 30 s resolution. The origin coordinates for the design of the domains were 22.896 N and 101.43 W, using a Lambert projection. The Tropical Physics Suite option was considered for running the WRF model.

The grid domains correspond to the different spatial scales of meteorological phenomena impacting the Veracruz port area, including mesoscale features like tropical cyclones and "nortes" and the sea breeze circulation.

WRF outputs were processed on the CALWRF processor to convert a specific file for CALMET (a module of CALPUFF) to reads hourly surface observations of wind speed, wind direction, temperature, cloud cover, ceiling height, surface pressure, relative humidity, and precipitation types of code.

## 4. Results and Discussion

### 4.1. Atmospheric Emissions

The $SO_2$ atmospheric emissions trend (Mg/day) from 2018 to 2020 in "Bahía Sur" is shown in Figure 5. In (a) it is presented the atmospheric emission corresponding to a sulfur content in marine fuel of 3.5% m/m from 2018 to 2019, while in 2020, a sulfur content of 0.5% m/m was used according to the IMO recommendations. On average, 5.2 and 5.3 Mg $SO_2$ were emitted daily in 2018 and 2019, respectively. In 2020 on average, 607 kg of $SO_2$ were emitted per day. The reduction from 2019 to 2020 was 88%.

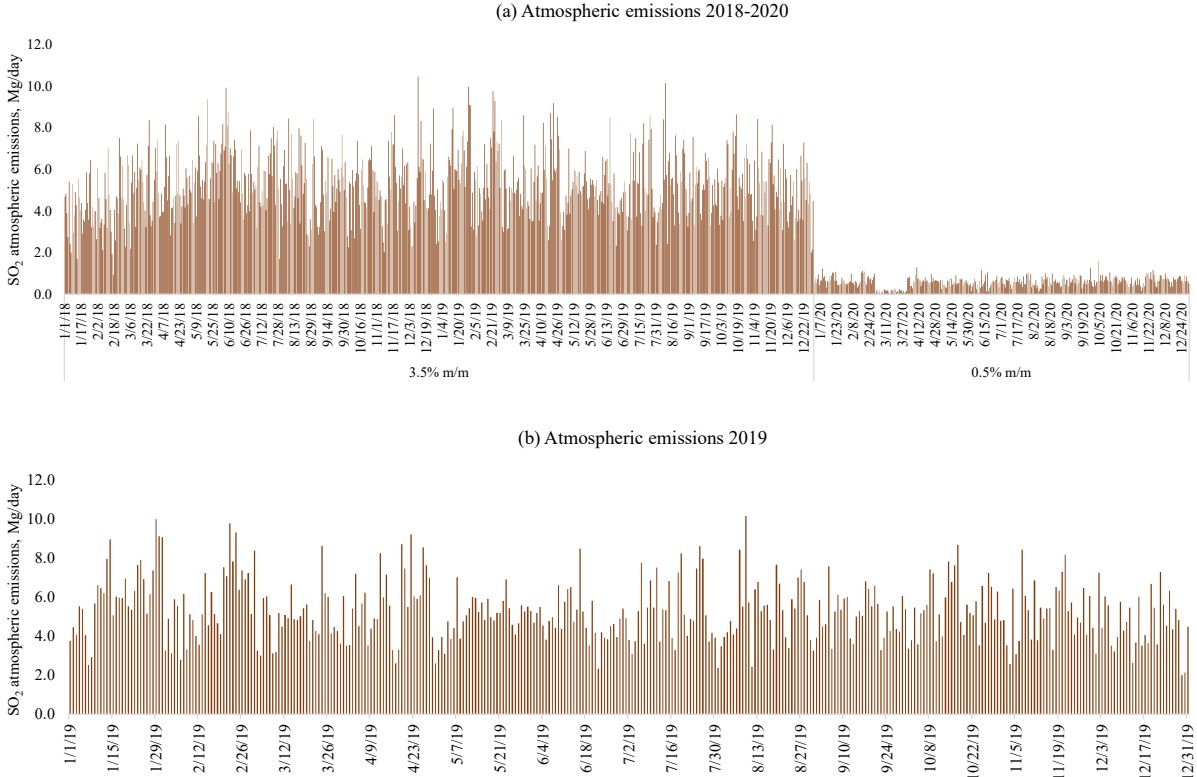

**Figure 5.** Temporal $SO_2$ atmospheric emissions by day for (**a**) 2018 to 2020, and (**b**) 2019.

The period considered for our results is shown in (b), as well as the distribution of atmospheric emissions, indicating that the operating cycle of the port system concerning ship movement is similar to 2018 and 2020.

During the COVID-19 pandemic, there was a reduction in the level of atmospheric emission in March 2020 due to the reduction in maritime traffic because of the pandemic controls. However, port activity was one of the sectors that did not have a particular restriction, observing that after March, maritime life continued steadily in the port of Veracruz.

*4.2. Air Quality and Meteorology*

The daily average $SO_2$ concentration ($\mu g/m^3$) and the wind speed (m/s) records and patterns from 2018 to 2020 are presented in Figure 6. From the observations in (a), it is possible to identify two distinct periods, November to May and June to October. From November to May, $SO_2$ concentrations were maximum of 2 $\mu g/m^3$ to 16 $\mu g/m^3$, and (b) wind speeds of 2 m/s to 14 m/s (50 km/h), indicating that the essential activities in the port are suspended due to the presence of meteorological phenomena. From June to October, the wind speeds were approximately 2 m/s, and $SO_2$ concentrations were minimal and constant. This information shows that it is necessary to evaluate the study period from November to May, and it is important to analyze the wind speed and $SO_2$ concentrations by component.

$SO_2$ analysis with respect to the meteorological data was performed using $SO_2$ temporal profiles, wind speed, and direction from AQMS. Figure 7a shows the highest $SO_2$ concentration occurred when the wind came from an ESE direction with 4 m/s. The wind speed is highest in the NNW, N, and NNE directions of 5 m/s to 9 m/s. The lowest concentrations were measured with the wind coming from NE to WNW, with speeds of approximately 2 m/s. According to the database from AQMS (Figure 7b), over 1000 were N (frequency of 14%) and W (frequency of 14%); however, the $SO_2$ concentrations were minimal (1.5 $\mu g/m^3$) in those directions. There is no significant contribution from S

components. The E and ESE directions occurred 10% of the time and ranged from 4 $\mu g/m^3$ to 5.5 $\mu g/m^3$ $SO_2$ concentrations prevail in those components.

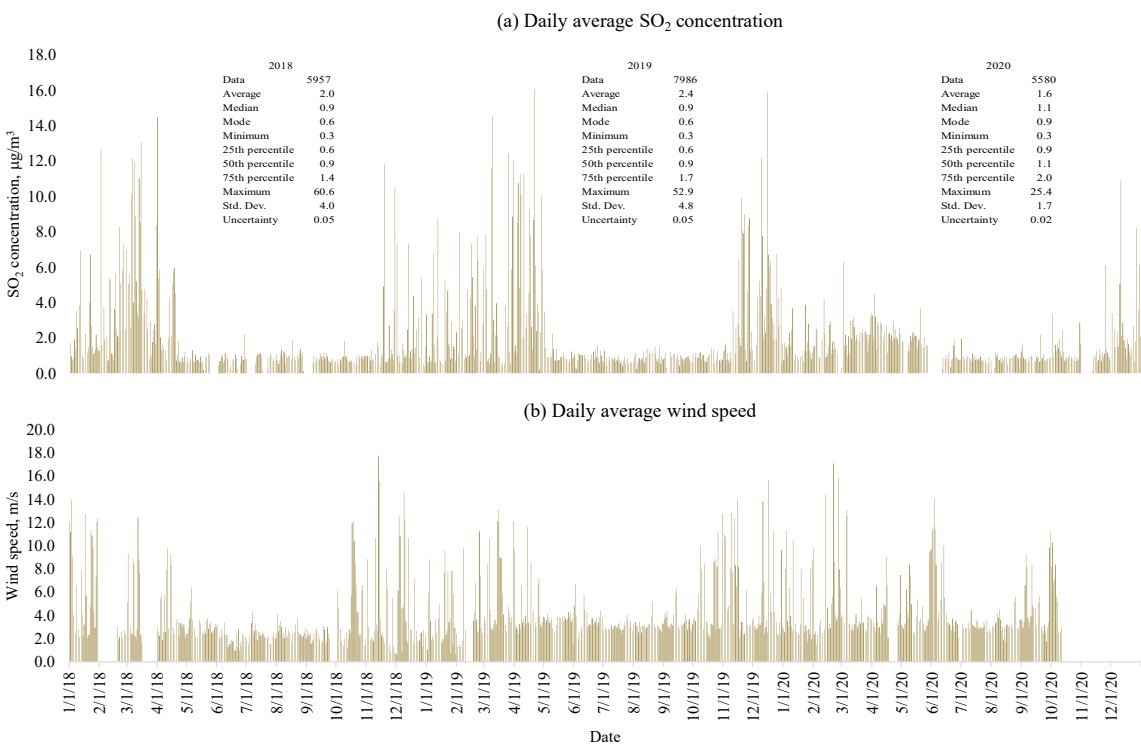

**Figure 6.** AQMS recording for (**a**) $SO_2$ ambient concentration and (**b**) wind speed from 2018 to 2020.

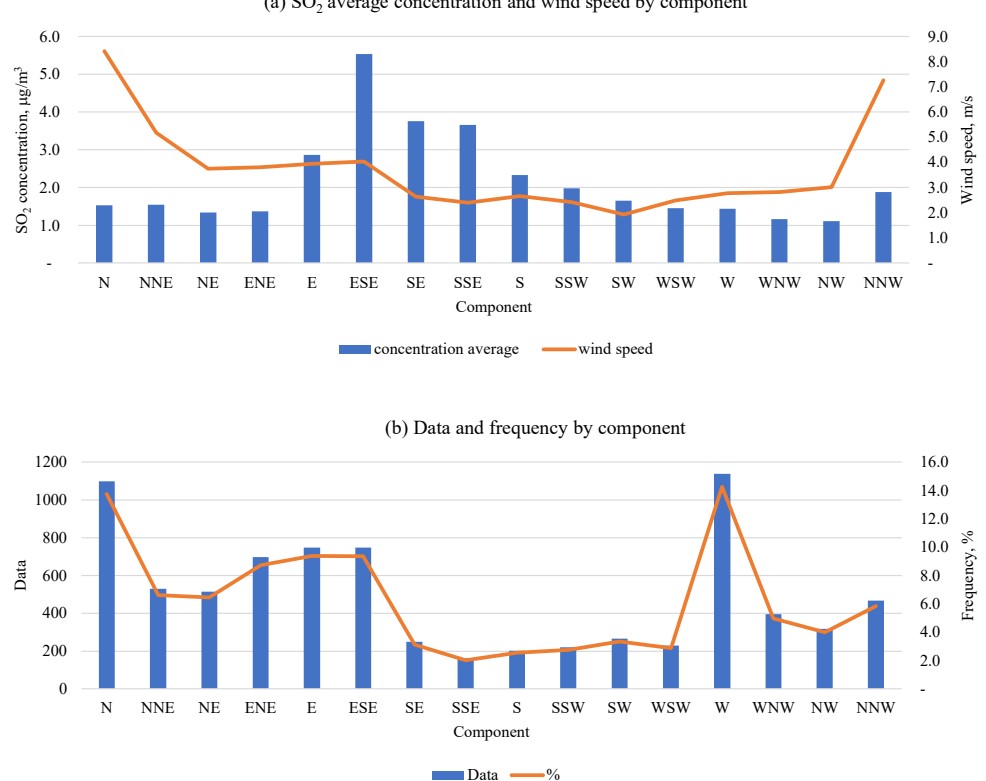

**Figure 7.** Air quality information for (**a**) $SO_2$ concentration and wind speed, and (**b**) data and frequency by component.

The SO$_2$ distribution for each season according to wind speed and direction is shown in Figure 8. The highest SO$_2$ concentrations were observed during Spring 2019, with SE winds. The frequency was 45%. Given this consideration, we decided to evaluate the air quality during this season. Also, Autumn and Winter represented an important contribution of SO$_2$ to AQMS, but the wind speed frequency is minimal in those seasons.

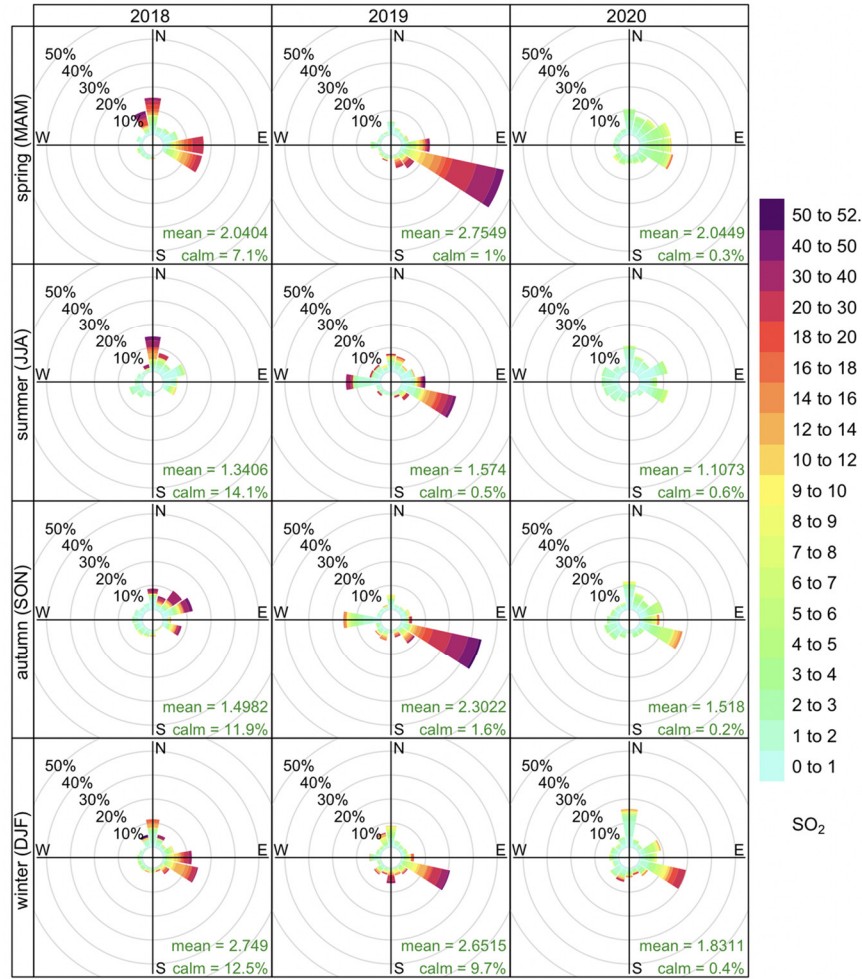

**Figure 8.** Prevalence of SO$_2$ concentration (µg/m$^3$) corresponding to AQMS.

The greatest activity within the port of Veracruz is in "Bahía Sur" since the "Bahía Norte" is being expanded. The "Bahía Sur" represented approximately 70% of the SO$_2$ contribution due to the movement of ships, evident when the wind blew from E to W. When northerly winds with speeds in excess of 40 km/h occur, activities in the port are suspended, indicating that the SO$_2$ contribution is not really from the movement of ships but from other emission sources outside the port.

### 4.3. Modeling of Air Quality and Meteorology

Backward trajectories (BackTraj) arriving at the AQMS site at an altitude of 10 masl, extending 120 h upwind, are shown in Figure 9. A cluster analysis was performed to process the meteorological database (2010 to 2020). The BackTraj's most frequent origin comes from the E, with frequencies of 26% (Cluster 1) and 38% (Cluster 2). The pattern of (a) indicates that air mass recirculation occurs [122] due to the constant interaction between air masses from N to E coverage and Yucatan Peninsula. Cluster 2 is consistent with the trade winds with air masses crossing the Yucatan Peninsula towards the study area. These air masses appeared from June to October and contributed to the transport of atmospheric pollutants from that area classified as industrial over the Gulf of Mexico. Trajectory clusters

3 and 4 represented northerly transport with frequencies of 10% and 26%, respectively. Cluster 3 is most frequent from December to May, while the slower-moving cluster 4 has a greater contribution during November and December. There is a clear trend in which the air masses were transported from N to S during the cold period, and from E to W in the dry period. This transport pattern is consistent with the five-year analysis of atmospheric transport to the El Tajín archaeological site, located 200 km to the NW of the Veracruz port [123].

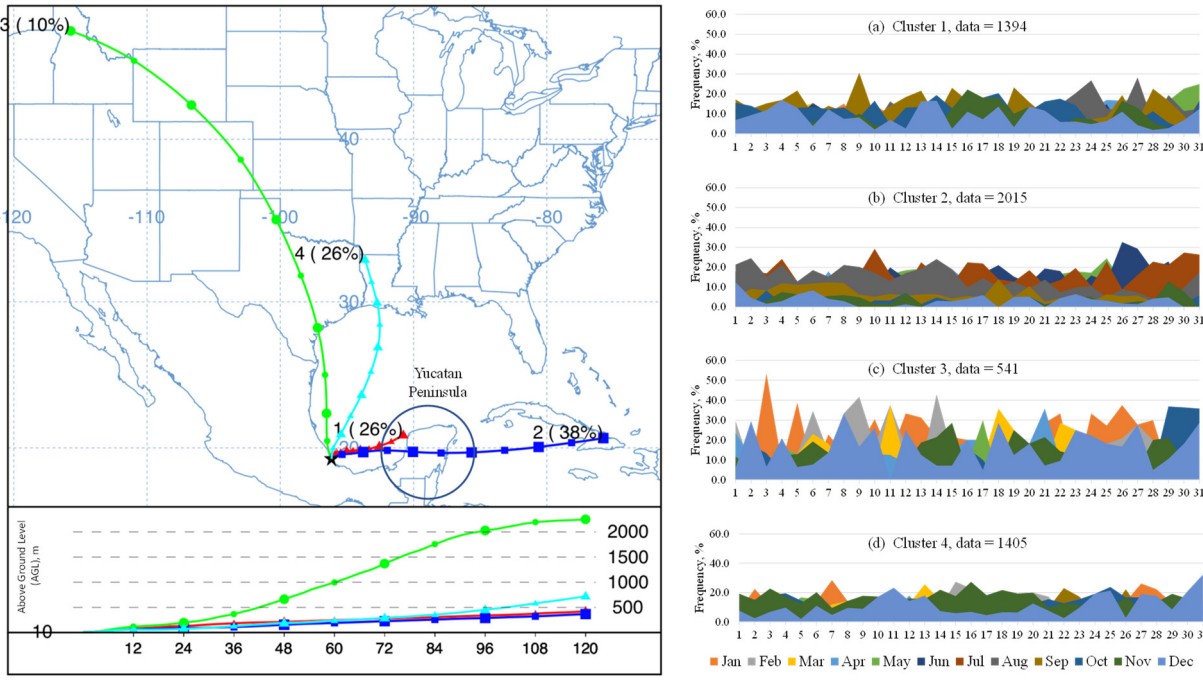

**Figure 9.** Backward trajectories from 2010 to 2020 and 120 h trajectory span.

Based on $SO_2$ monitoring observations, the wind rose, pollution rose, and BackTraj results, a set of days were selected for the air quality simulation (Table 3). There were 12 representative days, which occurred in March with a wind speed of 2 to 4 m/s from the maritime activity of "Bahía Sur". Therefore, the maximum concentration level did not occur at AQMS.

**Table 3.** Selected days to identify $SO_2$ ambient air concentration using the CALPUFF model.

| Event | Day | Air Quality Monitoring Station | | | Component | Bahía Sur Emissions, kg/day |
|---|---|---|---|---|---|---|
| | | $SO_2$ Concentration, µg/m³ | Wind Speed, m/s | Wind Direction, Degrees | | |
| 1 | 8/3/19 | 11.6 | 3.3 | 90.0 | East | 3111 |
| 2 | 9/3/19 | 14.5 | 2.6 | 90.0 | East | 3186 |
| 3 | 25/9/19 | 12.4 | 2.7 | 112.5 | East-southeast | 6187 |
| 4 | 30/3/19 | 12.0 | 2.8 | 112.5 | East-southeast | 3618 |
| 5 | 4/4/19 | 10.7 | 1.9 | 112.5 | East-southeast | 7184 |
| 6 | 6/4/19 | 11.3 | 2.5 | 112.5 | East-southeast | 5670 |
| 7 | 7/4/19 | 10.1 | 2.8 | 112.5 | East-southeast | 6212 |
| 8 | 10/4/19 | 11.3 | 3.0 | 112.5 | East-southeast | 4889 |
| 9 | 11/4/19 | 9.0 | 2.0 | 112.5 | East-southeast | 4875 |
| 10 | 16/4/19 | 9.2 | 2.2 | 112.5 | East-southeast | 3275 |
| 11 | 21/4/19 | 16.1 | 1.9 | 112.5 | East-southeast | 5488 |
| 12 | 28/4/19 | 10.0 | 2.2 | 112.5 | East-southeast | 6967 |

The emissions utilized in CALPUFF are shown in Table 4. The level of atmospheric emission was characterized by ship type corresponding to the destination dock; that is,

each ship represented a point source of atmospheric emission. The number of ships per day influences the atmospheric emission volume, as well as the ship type. It was found that Container emitted the highest emissions followed by Liquid. The first is due to its GT size with stay times of approximately 20 h, while Liquid was lower in GT, but docking time was greater than 50 h for the selected days.

**Table 4.** Atmospheric emission (kg/day) by type of vessel for the days selected in CALPUFF.

| Dock | Type | Days | | | | | | | | | | | |
|------|------|--------|--------|---------|---------|--------|--------|--------|---------|---------|---------|---------|---------|
| | | 8/3/19 | 9/3/19 | 25/3/19 | 30/3/19 | 4/4/19 | 6/4/19 | 7/4/19 | 10/4/19 | 11/4/19 | 16/4/19 | 21/4/19 | 28/4/19 |
| 1 | General, RoRo Cargo | - | 287 | 870 | 45 | 45 | 45 | - | 90 | 742 | 147 | 1186 | - |
| 2 | General | 57 | 57 | 114 | 61 | 92 | 98 | 355 | 256 | - | 125 | 229 | - |
| 4 | General, Dry Bulk | 281 | 281 | 301 | 49 | 516 | 934 | 934 | 387 | 431 | 44 | 498 | 451 |
| 5 | General, Dry Bulk | 92 | - | - | 301 | - | - | - | 562 | 562 | 289 | 1240 | - |
| 6 | Dry Bulk | - | - | 553 | 320 | 1246 | 89 | 89 | 89 | 241 | 758 | 617 | 771 |
| 7 | Container, RoRo Cargo | - | 276 | 1361 | 361 | - | 2231 | 1240 | 48 | 611 | 235 | 684 | 1659 |
| 8 | Dry Bulk | 569 | 569 | 206 | 682 | 469 | 469 | 681 | 469 | 218 | - | 393 | 545 |
| 9 | Cement | 354 | 354 | - | - | 271 | 538 | 703 | 142 | 745 | - | 343 | 569 |
| 11 | Container | 1346 | 338 | 1090 | 972 | 2922 | 338 | 972 | 2124 | 916 | 1078 | - | 1907 |
| 16 | Liquid Bulk | 412 | 1025 | 1692 | 827 | 1624 | 927 | 1238 | 721 | 410 | 600 | 299 | 1064 |
| | Total | 3111 | 3186 | 6187 | 3618 | 7184 | 5670 | 6212 | 4889 | 4875 | 3275 | 5488 | 6967 |

The $SO_2$ concentration from the CALPUFF model is shown in Table 5 for each reference case. These results compare AQMS and the air quality simulation and do not indicate the maximum concentration level observed at the port. The maximum atmospheric emission levels do not always indicate that there will be a high concentration or vice versa, as it depends on the atmospheric stability of the day, cloud cover, solar radiation, wind speed and direction, and type of thermal inversion during the day, among other factors.

**Table 5.** $SO_2$ ambient air concentration ($\mu g/m^3$) using the CALPUFF model.

| Event | Day | Wind Speed, m/s | Wind Direction, Degrees | Wind Component | Bahía Sur Emissions, kg/day | AQMS $\mu g/m^3$ | Modeling $SO_2$ Concentration at 24 h Average, $\mu g/m^3$ | | | | |
|-------|-----|-----|------|----------------|------|-------|-------|-------|-------|-------|-------|
| | | | | | | | [103] | [104] | [105] | [106] | [107] |
| 1 | 8/3/19 | 3.3 | 90.0 | East | 3111 | 11.61 | 10.30 | 6.80 | 5.94 | 12.10 | 11.60 |
| 2 | 9/3/19 | 2.6 | 90.0 | East | 3186 | 14.55 | 7.70 | 6.70 | 6.00 | 7.70 | 7.90 |
| 3 | 25/3/19 | 2.7 | 112.5 | East-southeast | 6187 | 12.42 | 7.80 | 7.40 | 6.90 | 8.20 | 9.10 |
| 4 | 30/3/19 | 2.8 | 112.5 | East-southeast | 3618 | 12.04 | 13.80 | 12.80 | 11.79 | 15.50 | 16.90 |
| 5 | 4/4/19 | 1.9 | 112.5 | East-southeast | 7184 | 10.74 | 14.00 | 10.60 | 9.10 | 18.40 | 14.20 |
| 6 | 6/4/19 | 2.5 | 112.5 | East-southeast | 5670 | 11.27 | 6.00 | 6.00 | 5.83 | 6.10 | 7.00 |
| 7 | 7/4/19 | 2.8 | 112.5 | East-southeast | 6212 | 10.08 | 10.90 | 8.20 | 8.45 | 12.00 | 12.80 |
| 8 | 10/4/19 | 3.0 | 112.5 | East-southeast | 4889 | 11.32 | 6.00 | 7.10 | 6.88 | 7.70 | 8.80 |
| 9 | 11/4/19 | 2.0 | 112.5 | East-southeast | 4875 | 9.02 | 7.30 | 6.20 | 6.94 | 7.50 | 7.50 |
| 10 | 16/4/19 | 2.2 | 112.5 | East-southeast | 3275 | 9.18 | 10.50 | 9.20 | 9.74 | 11.40 | 10.80 |
| 11 | 21/4/19 | 1.9 | 112.5 | East-southeast | 5488 | 16.10 | 16.80 | 18.00 | 16.46 | 19.00 | 20.00 |
| 12 | 28/4/19 | 2.2 | 112.5 | East-southeast | 6967 | 9.99 | 23.00 | 20.20 | 16.80 | 22.40 | 24.50 |

The $SO_2$ ambient concentration distribution using the CALPUFF model is shown in Figure 10. It was found that $SO_2$ concentration was similar for [103–107] but differed from the AQMS because there are other non-ship emission sources not included in the CALPUFF simulations. Likewise, the atmospheric stability changes during the day, while the model considers this parameter to be constant. Nevertheless, a similarity was found between the results obtained through the CALPUFF model and the record of the environmental monitoring station.

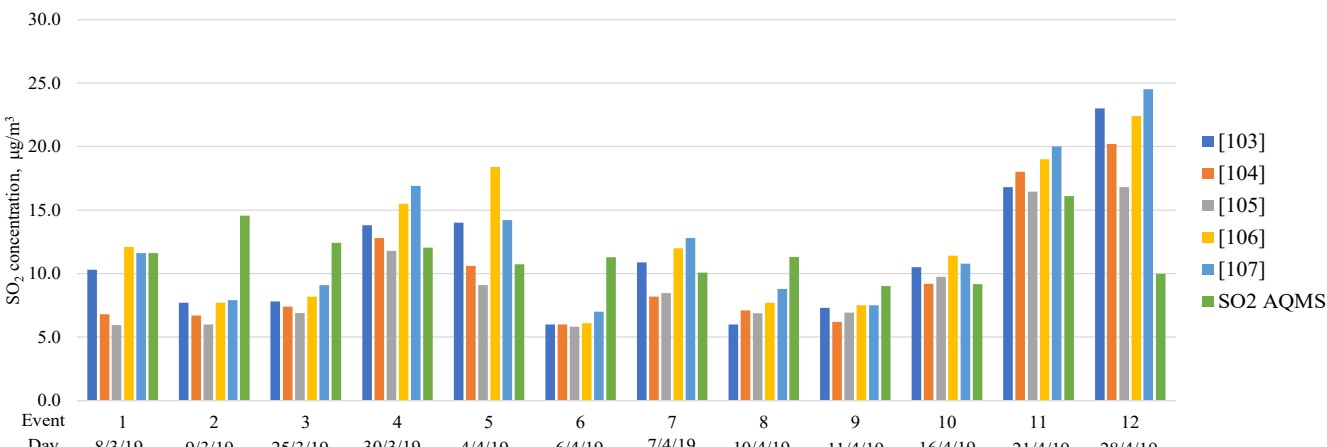

**Figure 10.** Distribution of SO$_2$ concentration (µg/m$^3$) between the CALPUFF model and the AQMS.

The SO$_2$ maximum concentrations from CALPUFF are shown in Figure 11. It was found that the 24 h average concentration limit established by [112,113] for SO$_2$ was exceeded on five occasions. Using the emissions from [105], however, the reference was never exceeded. The concentration using information from [106] provided a higher concentration level than [103] due to the different specifications of the vessels. The results found for [104] were similar to [103].

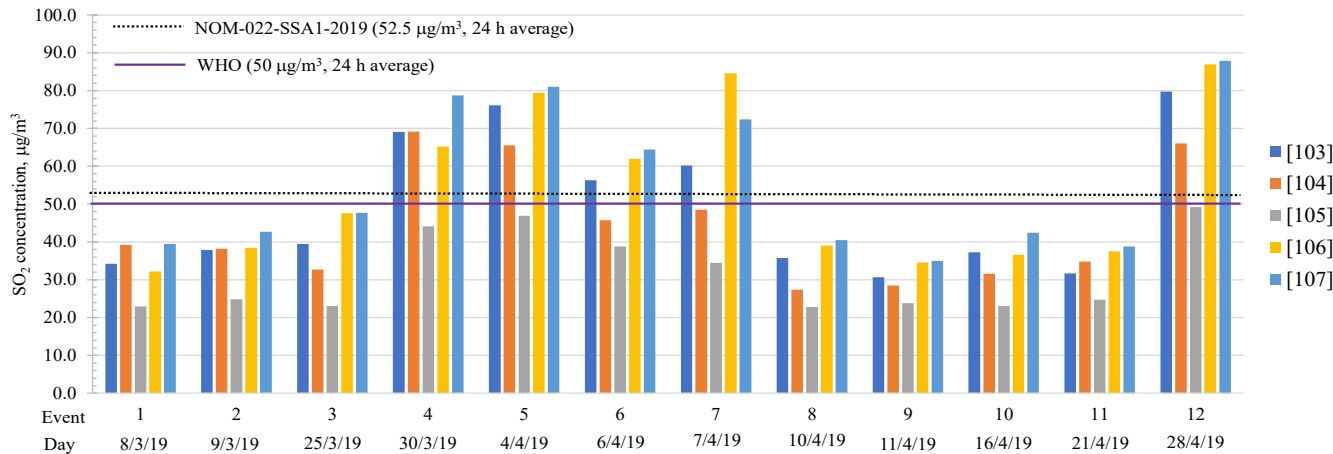

**Figure 11.** Maximum SO$_2$ concentration using the CALPUFF model.

An example of the impact of port activities on air quality, according to the CALPUFF model, is shown in Figure 12 for day 4/28/19. On this day, the SO$_2$ concentration exceeded the 24 h standard average for (a) [103], (b) [104], (c) [106], and (d) [107]. It was found that the maximum concentration was 0.5 km from the emission source; thus, the greatest impact prevails inside the port area and affects the city of Veracruz, located SW of the port. Outside a 3 km radius of influence, the concentration is lower due to atmospheric dispersion, chemical destruction, and deposition.

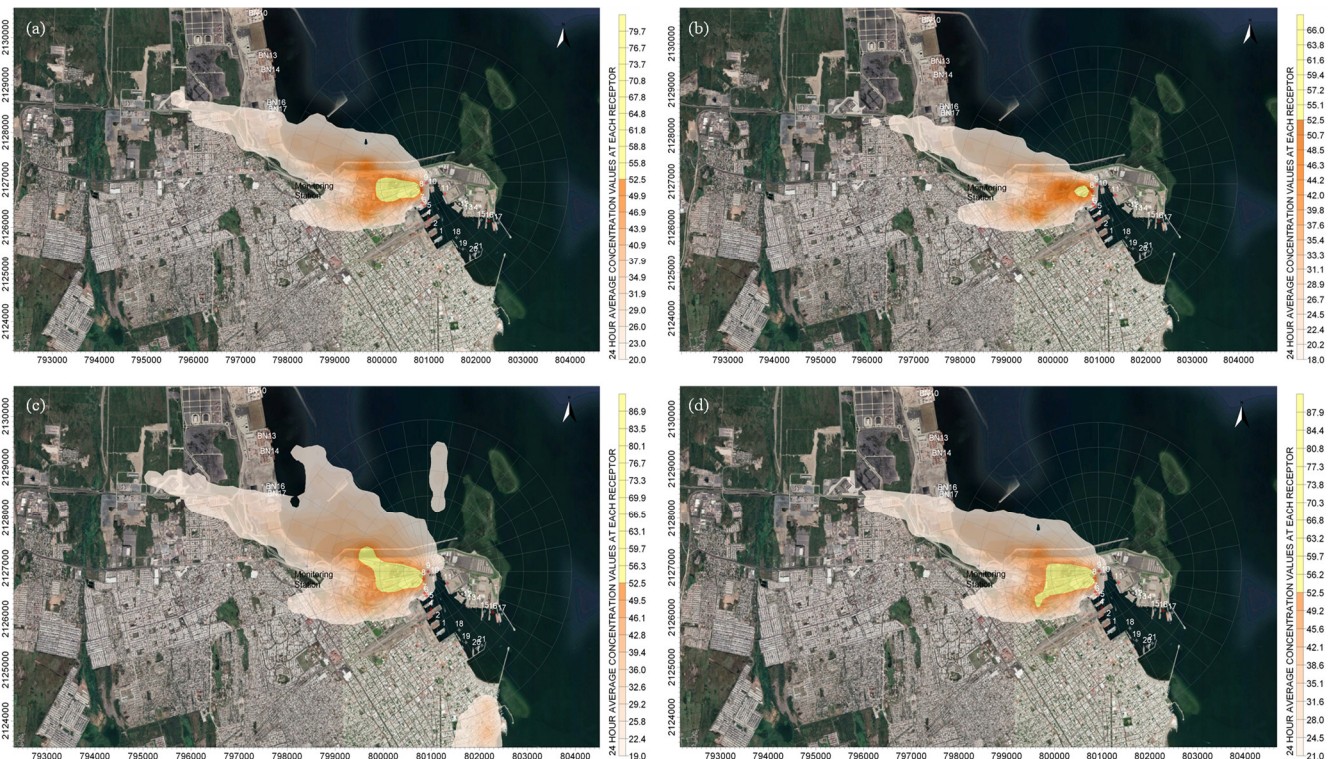

**Figure 12.** Impact on air quality in the city of Veracruz according to the CALPUFF model for (**a**) [103], (**b**) [104], (**c**) [106], and (**d**) [107].

Finally, the 24 h average $SO_2$ reference concentration update (40 µg/m$^3$ as an air quality guideline and 50 µg/m$^3$ as an interim target) recommended by [112] is very strict and may not be practical due to regional differences in anthropogenic activities. It is impossible to regulate a global reference concentration because there are countries in development that will continue to use fossil fuels with high sulfur content. However, it is necessary to improve methods to prevent, minimize, and control atmospheric emissions to guarantee satisfactory air quality.

## 5. Conclusions

Marine fuel quality implementation of 0.5% m/m significantly reduced the $SO_2$ atmospheric emission by 88% from 2019 to 2020.

Ambient air quality from 2018 to 2020 was satisfactory according to the location of the air quality monitoring station.

The ambient air quality monitoring location did not record the maximum concentration because the distance from the port to the monitoring station was 5 km, and it is important to assess the impact under 500 m. Its location made it possible to identify the presence of a sea breeze most of the time, as well as the $SO_2$ concentration record when the wind blew from N to S.

It is necessary to monitor air quality compliance 500 m southwest of the port from November to May due to a combination of high $SO_2$ concentration and high occurrence of this wind direction.

The maximum concentration occurred 500 m downwind of the port, with ambient $SO_2$ concentration exceeding the 24 h Mexican Air Quality Standard continuously for four days, according to air quality modeling. The combination of a shallow planetary boundary layer (200 to 400 m), low wind speed (<3 m/s), and atmospheric emissions (3.6 to 7 Mg/day) caused events of poor air quality.

## 6. Recommendations

Establish an environmental monitoring network of automatic analyzers based on the equivalent method for ozone ($O_3$), nitrogen oxides ($NO_x$), black carbon (BC), carbon monoxide (CO), and particles (PM) to strengthen the record of ambient air.

Consider other, non-ship sources of atmospheric emissions from the port of Veracruz and integrate them into the environmental pollution model to identify their contribution.

## 7. Future Work

Applying the Chemical Transport Model (CTM) to identify interactions with ozone and solar radiation in the area considering the presence and absence of ship movement.

**Author Contributions:** Conceptualization, G.F.G., R.S.E. and A.G.R.; Data curation, J.M.B.R., A.G.R., R.S.E., V.M.R. and J.D.W.K.; Formal analysis, A.R.H., G.F.G. and J.M.B.R.; Investigation, G.F.G., R.S.E. and A.G.R.; Methodology, G.F.G., R.S.E. and J.M.B.R.; Resources, G.F.G.; Supervision, R.S.E., A.G.R. and J.M.B.R.; Validation, G.F.G., A.R.H., J.M.B.R. and V.M.R. and J.D.W.K.; Software, G.F.G., V.M.R. and A.G.R.; Writing-review and editing, J.D.W.K. All authors have read and agreed to the published version of the manuscript.

**Funding:** This research did not receive any specific grant from funding agencies in the public, commercial, or not -for-profit sectors.

**Institutional Review Board Statement:** Not applicable.

**Informed Consent Statement:** Not applicable.

**Data Availability Statement:** Not applicable.

**Acknowledgments:** The authors of this study are grateful for the participation of José Hernández Téllez and Humberto Bravo Witt respect to the management of air quality monitoring station in port of Veracruz. We are grateful of the staff of Sección de Contaminación Ambiental at ICAyCC–UNAM: Ana Luisa Alarcón Jiménez, Pablo Sánchez Álvarez, Elizabeth Vega Rangel, Elías Granados Hernández, Rafael Antonio Durán, Mauro Cortez Huerta. Cómputo y Súper Cómputo at ICAyCC: Dulce Rosario Herrera Moro, Pedro Damián Cruz Santiago. Instituto de Geografía de la UNAM: Gustavo Vázquez Cruz. Participation of the staff of Administración Portuaria Integral de Veracruz (APIVER): Francisco Lianó C., Socaris de la Luz, and David Augusto de la O N. Agreement between APIVER and UNAM: "Evaluación de la calidad del aire, depósito atmosférico y meteorología para desarrollar el programa para la prevención y minimización del posible deterioro ambiental significativo en el Recinto Portuario de Veracruz y en las zonas de interés". Finally, we acknowledge Instituto de Ciencias de la Atmósfera y Cambio Climático for supporting the postdoctoral appointment of G.F.G., at Universidad Nacional Autónoma de México.

**Conflicts of Interest:** The authors declare no conflict of interest.

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
