# Peer review of "Sea Port SO2 Atmospheric Emissions Influence on Air Quality and Exposure at Veracruz, Mexico"

_atmosphere, doi:10.3390/atmos13121950_

Round 1

Reviewer 1 Report

The comments and suggestions are reported in the attached file.

Author Response

Dear reviewer 1

We appreciate your spent time to review our study “Sea port SO2 emissions influence in air quality and exposure. Case study: Veracruz, Mexico”. This topic is very interesting in Mexico because the marine activity is growing in the Gulf of Mexico, and it is necessary to reinforce the atmospheric emissions inventories and air quality simulation in ports. Atmospheric emission in addition air quality, meteorology and modeling were evaluated to identify the exposure level over the city of Veracruz due to the movement of ships. According to your observations, we have responded your questions and comments. We are convinced that your comments will benefit our study to meet the necessary requirements to be published in atmosphere journal. The English language and style were carried out by our co-worker Jonathan Kahl of University Wisconsin-Milwaukee at United States (USA).

Section 1 Introduction

Comment:

The introduction and background provide a good state of the art and include relevant references, but the content of the two paragraphs is closely related and, in some case, redundant. I suggest merging these two paragraphs in one.

Response:

Thank you for your suggestion. We have added the Background section into Introduction section. The change you can see in this new review. Also, one more reviewer indicated the same idea.

Section 3 Methodology

Comment:

The section of methodology should be divided into subparagraphs. Also, data collection could be joint into “Atmospheric emission”.

Response:

Thank you for your observation. We have added the subparagraphs for this new review to indicate what kind of information we used to develop this study. Data collection section was moved into atmospheric emission section due to this information is corresponding to it.

Comment:

Lines 238-239 “from August 2017 to December 2020”. If the period of interest is from 2018 to 2020, Why is the meteorology considered starts from August 2017?

Response:

Thank you for your question. For us was interesting to integrate the complete records form air quality monitoring station from 2017 to 2020 by SO2 and meteorology. However, our broadcast starts in 2018 to estimate the atmospheric emissions. For this, we have decided to remove the period from August to December 2017 when the air quality monitoring station started the record of air quality and meteorology. In this case, from 2018 to 2020 is a good period to characterize and evaluate the wind speed and wind direction for the port of Veracruz. These changes you can see in Figure 2 (wind rose plot) and Figure 8 (pollutant rose plot) for this new review.

Comment:

Lines 164-165: “the atmospheric emissions estimation on daily basis”. The authors should justify the choice of the calculation of emission on daily basis and why they haven’t estimate the emissions on hourly basis.

Response:

Thank you for your comment. Daily official information about the movement of ships is through from port of Veracruz Administration. This official information is published each month. Given this consideration, we can estimate the atmospheric emissions on daily basis. However, to estimate the atmospheric emission on hourly basis it is necessary to disaggregate the atmospheric emissions considering the time spent in port (mooring, berthing, and maneuvering positions). Currently, spent time in berthing and maneuvering position are known in specific, but in mooring position there is not known in specific. For this, we have estimated the atmospheric emissions on daily basis, and integrate it in CALPUFF to identify the SO2 concentration, transport and dispersion over Veracruz city.

Comment:

Lines 198-199. The ship information as well as the Gross Tonnage, ME and AE, are one of the main determinants in the calculation of emissions and this information of each ship is different. For this reason, the authors should be report a table with the ship information used in their work.

Response:

Thank you for your suggestion. You can see the GT, power of ME and AE distribution in Figure 3 as you suggested. These parameters were calculated according to the typology of ship of the port from 2018 to 2019 average. This information represents a complement with the ship information used in our study.

Comment:

Line 305. It would be useful to specify: the meteorological parameters (stability class 26) considered during the simulation of wind fields with WRF, the coupling conditions of WRF and CALMET, the 3D wind field used for CALMET initialization, which options were retained for the prognostic model?

Response:

Thank you for your recommendation. We have added that WRF outputs were processed on CALWRF processor to convert a specify file for CALMET (a module if CALPUFF) to reads hourly surface observations of wind speed, wind direction, temperature, cloud cover, ceiling height, surface pressure, relative humidity, and precipitation types of code. The Global Tropospheric Analysis and Forecast Grid were used. Parameters include surface pressure, sea level pressure, geopotential height, temperature, sea surface temperature, soil values, ice cover, relative humidity, u- and v- winds, vertical motion, vorticity, and ozone. This information you can see in the 3.2. section: Air quality modeling and meteorology for this new review.

Section 4 Results

Comment:

Lines 318-319: “The SO2 atmospheric emissions trend (Mg/day) from 2018 to 2019 in “Bahía Sur” is 318 shown in Figure 3”. Figure 3 shows the concentration from 2018 to 2020.

Response:

Currently, this Figure is number 5 (in this new review). Description in Figure 5 is corresponding to atmospheric emissions considering to 3.5% mass by mass (previously 2020) and 0.5% mass by mass (after 2020) of sulfur content in marine fuel according to the International Maritime Organization. In a) is shown the atmospheric emissions distribution from 2018 to 2020, and b) is shown only 2019, period considered to air quality simulation.

Comment:

Lines 356-357. “The hourly average SO2 concentration (μg/m3) and the wind speed (m/s) records and patterns during 2017 to 2020 are presented in the Figure 4”. The authors have estimated the daily SO2 emissions for this reason it’s more appropriate report the daily average SO2 concentration recorded in AQMS

Response:

Thank you for your suggestion. Currently, this Figure is number 6 (in this new review). Taking care of your comment, we have presented the SO2 concentration daily average as well as the wind speed according to the records from air quality monitoring station from 2018 to 2020 (period of 2017 has removed according to Point 2.1)

Comment:

Figure 5. The figures 5c and 5d are unclear. The authors could represent the same information with pollution rose and wind rose, respectively.

Response:

Thank you for your observation. Also, one more reviewer has indicated the same idea. For this new review, we have modified the period (2018 to 2020) considering to SO2 concentration daily average, and we removed Figure 5c and 5d because they integrated hourly information, and it is not corresponding to daily average. You can see it in Figure 7.

Minor revisions

Comment:

Line 181: What do the authors mean by ET?

Response:

Thank you for your observation. This abbreviation means the total atmospheric emission (berthing and maneuvering positions). However, we have modified it (in this new review) because it is confused to understand. For this,  was modified by “E” in Equation 1.

Comment:

Line 230: The reference to the Figure 1 is necessary.

Response:

Thank you for your suggestion. We have referenced the Figure 1 for this new review. This Figure was designed by CALPUFF model using its WEBGIS for land use and topography to port of Veracruz. We used the latitude and longitude to locate the docks for the port (Google Earth), and official information from the port captaincy.

Comment:

Figure 1: To better understand the figure, I suggest to split the Figure 1 in two figures.

Response:

Thank you for your recommendation. To better understand we have modified the Figures 1, 2 and 8. These Figures are clearer and have a better representation for their description. You can see it in this new review.

Reviewer 2 Report

While this study seems promising and scientifically sound, the extensive need for English language proof-reading and also figure formatting makes it difficult at times to really understand what the authors are trying to convey. This makes it challenging for me to provide a comprehensive review of the manuscript because it is likely that I did not fully grasp some points raised by the authors. I have included a few comments below, but ideally I would like to revisit this manuscript once it has been proof-read for English and data visualization clarity. However, I understand that this is my limitation, and so if other reviewers are able to provide their review without this issue, the manuscript should be considered for publication based on their review more.

Specific comments:

- Abstract should be a bit longer, describing the results more.

- The background and introduction sections seem repetetive, and should be merged to form a single introduction section.

- Eq. 1: the first line of Eq. 1 seems to define the variable E, while the second line defines E_T, even though these both seem related. Please correct/clarify.

- Fig. 5C and D are hard to read, and might have some error. The orange “wind direction” trace looks like a cumulative distribution. The discussion states from this figure that “when the wind blows from W direction, the concentrations were below 5 ug/m3”, but this is not readily apparent from the figure. It is also unclear what the x-axis labels mean – this should ideally be a time series, but it seems to not be the case. I think the information in these charts should be displayed using a wind rose or pollutant rose diagrams, similar to Fig. 1 and 7.

Thank you for considering me for this review, and I apologize that I cannot provide a more helpful review at this time.

Author Response

Dear reviewer 2 

We appreciate your spent time to review our study “Sea port SO2 emissions influence in air quality and exposure. Case study: Veracruz, Mexico”. This topic is very interesting in Mexico because the marine activity is growing in the Gulf of Mexico, and it is necessary to reinforce the atmospheric emissions inventories and air quality simulation in ports. Atmospheric emission in addition air quality, meteorology and modeling were evaluated to identify the exposure level over the city of Veracruz due to the movement of ships. According to your observations, we have responded your questions and comments. We are convinced that your comments will benefit our study to meet the necessary requirements to be published in atmosphere journal. The English language and style were carried out by our co-worker Jonathan Kahl of University Wisconsin-Milwaukee at United States (USA).

Comment:

Abstract should be a bit longer, describing the results more.

Response:

Thank you for your comment. Following your recommendation, we have modified the Abstract including our findings according to the results obtained from this study. You can note it in this new review.

Comment:

The background and introduction sections seem repetitive and should be merged to form a single introduction section.

Response:

Thank you for your comment. One more reviewer indicated the same idea. For this, we have moved the Background section into the Introduction section, and we are convinced that it represents a better structure for the document.

Comment:

Eq. 1: the first line of Eq. 1 seems to define the variable E, while the second line defines E_T, even though these both seem related. Please correct/clarify.

Response:

Thank you for your comment. This is a mistake by us. The  abbreviation means the total atmospheric emission (in berthing and maneuvering positions), but we have corrected this mistake, that is, we have changed  to E (total atmospheric emission). This correction you can see in the Equation 1 for this new review.

Comment:

Fig. 5C and D are hard to read and might have some error. The orange “wind direction” trace looks like a cumulative distribution. The discussion states from this figure that “when the wind blows from W direction, the concentrations were below 5 ug/m3”, but this is not readily apparent from the figure. It is also unclear what the x-axis labels mean this should ideally be a time series, but it seems to not be the case. I think the information in these charts should be displayed using a wind rose or pollutant rose diagrams, similar to Fig. 1 and 7.

Response:

Thank you for your observation. According to another reviewers, we have modified the Figures content considering to 1) the period of study from 2018 to 2020, 2) only SO2 concentration daily average, and 3) clarifying the Figure 1, 2, and 8. Given this consideration, we removed Figure 5c and 5d because they integrated hourly information, and it not corresponding to daily average. You can see these changes in Figure 7. Figure 2 (wind rose plot) and Figure 8 (pollutant rose plot) represent a complement of Figure 7. It is necessary to comment you Figure 7a content is respect to the SO2concentration average by component of wind, and wind speed.

Reviewer 3 Report

The paper describes a a comprehensive methodology to model and analyse the environmental impact of vessels in a rather big port in terms of airborne pollutants.

The paper follows a well-known and established methodology and presents an interesting case study. It is well organized and comprehensive soit deserves publication. However, I would suggest the following amendements.

If possible, I would add an analysis of other airborne pollutants, primarily PM10 and PM2.5. Using the same methodology, it would be rather easy to check such effects.

English is poor and contains errors, mistakes, and verbosities. It badly needs a complete check possibly performed by a professional.

Most figures and graphs should be enlarged and better explained.

Conclusions are rather short and could be enhanced. Personally, I would include “Recommendations” and “Future work” within them.

In conclusion, the paper should be accepted with the minor suggested modifications. Possibly I would introduce and analysis of other pollutants.

Author Response

Dear reviewer 3

We appreciate your spent time to review our study “Sea port SO2 emissions influence in air quality and exposure. Case study: Veracruz, Mexico”. This topic is very interesting in Mexico because the marine activity is growing in the Gulf of Mexico, and it is necessary to reinforce the atmospheric emissions inventories and air quality simulation in ports. Atmospheric emission in addition air quality, meteorology and modeling were evaluated to identify the exposure level over the city of Veracruz due to the movement of ships. According to your observations, we have responded your questions and comments. We are convinced that your comments will benefit our study to meet the necessary requirements to be published in atmosphere journal. The English language and style were carried out by our co-worker Jonathan Kahl of University Wisconsin-Milwaukee at United States (USA).

Comment:

If possible, I would add an analysis of other airborne pollutants, primarily PM10 and PM2.5. Using the same methodology, it would be rather easy to check such effects.

Response:

Thank you for your suggestion and recommendation. The goal of this study was to assess the air quality impact over the city of Veracruz due to the movement of vessels considering to the typology of ship. The principal component of the air to assess the impact was SO2 which it is emitted during the combustion process for the main engine and auxiliary engine. Fossil fuels will continue to be used in Mexico and considering to the sulfur content is its main component, it is necessary to identify the impact on air quality. Therefore, in this study the influence of atmospheric emissions by SO2 was analyzed over the city of Veracruz, adding SO2 air quality, meteorology and air quality modeling.

We know that one fraction of sulfur content in marine fuel is corresponding to particles (PM) and is very important to assess the impact considering to the fraction for PM10 and PM2.5. The PM total emission, ~80% is corresponding to PM10, and ~95% for PM2.5 (from PM10). Estimation of PM, PM10 and PM2.5 were determined by Fuentes et al (2021). They founded from 2018 to 2019 an emission of 34 Mg/year for PM, 28 Mg/year of PM10, and 27 Mg/year of PM2.5.

We are convinced that the analysis between PM and SO2 is very interesting, however, we propose that this could be reflected in another study and deepen its analysis considering to meteorological phenomena that prevail in the port – city

Comment:

English is poor and contains errors, mistakes, and verbosities. It badly needs a complete check possibly performed by a professional.

Response:

Thank you for your observation. For this new review our co-worker Jonathan Kahl of University Wisconsin-Milwaukee at United States (USA) carried out the English language and style. The correct style of English made by our co-worker will improve the understanding of our study and its content.

Comment:

Most figures and graphs should be enlarged and better explained.

Response:

Thank you for your comment. For this new review, and according to another reviewers, we have modified the Figures content considering to 1) the period of study from 2018 to 2020, 2) only SO2 concentration daily average, and 3) clarifying the Figure 1, 2, and 8. Given this consideration, we removed Figure 5c and 5d because they integrated hourly information, and it not corresponding to daily average. The correct style of English made by our co-worker will improve the understanding of our study and its content. These modifications you can see in Figure 7 (in this new review).

Comment:

Conclusions are rather short and could be enhanced. Personally, I would include “Recommendations” and “Future work” within them.

Response:

Thank you for your suggestion. According to another reviewers, the structure of the document is good. They did not comment us about the recommendations and future work section. For this, we would like to keep the recommendations and future work sections because it is always necessary to suggest for improving in all aspects in a develop country. We hope that you can understand us about it.

Comment:

In conclusion, the paper should be accepted with the minor suggested modifications. Possibly I would introduce and analysis of other pollutants.

Response:

Thank you for your suggestion. The scope of this study was evaluating the impact on air quality considering to the influence of atmospheric emissions in addition air quality by SO2, meteorology and modeling. Our recommendations are aimed for monitoring other pollutants as ozone (O3), nitrogen oxides (NOx), black carbon (BC), particles (PM10, and PM2.5), and carbon monoxide (CO) to relate it with meteorological factors. However, our study reflects the SO2 panorama in Mexico because in next years will be continue use the fossil fuels, and the sulfur content in fossil fuel is approximately of 3.5% mass by mass.

---

Reference:

Fuentes, G. G., Baldasano, R. J. M., Sosa, E. R., Granados, H. E., Zamora, V. E., Antonio, D. R., Kahl, W. J., Estimation 672 of atmospheric emissions from maritime activity in the Veracruz port, Mexico. J. Air Was. Manag. Assoc. 2021, 71, 673 934-948. https://doi.org/10.1080/10962247.2021.1902421

Round 2

Reviewer 1 Report

I am satisfied with the author’s responses to my questions/issues raised in my initial review. The revised manuscript is easier to follow based on feedback from the reviewers.  I recommend that the revised paper be accepted in present form.